# Fear of Infection and Depressive Symptoms among German University Students during the COVID-19 Pandemic: Results of COVID-19 International Student Well-Being Study

**DOI:** 10.3390/ijerph19031659

**Published:** 2022-01-31

**Authors:** Franca Spatafora, Paula M. Matos Fialho, Heide Busse, Stefanie M. Helmer, Hajo Zeeb, Christiane Stock, Claus Wendt, Claudia R. Pischke

**Affiliations:** 1Institute of Medical Sociology, Centre for Health and Society, Medical Faculty, Heinrich Heine University Duesseldorf, 40225 Duesseldorf, Germany; franca.spatafora@med.uni-duesseldorf.de (F.S.); PaulaMayara.MatosFialho@med.uni-duesseldorf.de (P.M.M.F.); 2Department Prevention and Evaluation, Leibniz-Institute for Prevention Research and Epidemiology—BIPS, 28359 Bremen, Germany; busse@leibniz-bips.de (H.B.); zeeb@leibniz-bips.de (H.Z.); 3Institute of Health and Nursing Science, Charité—Universitätsmedizin Berlin, Corporate Member of Freie Universität Berlin and Humboldt-Universität zu Berlin, 10117 Berlin, Germany; stefanie.helmer@charite.de (S.M.H.); christiane.stock@charite.de (C.S.); 4Health Sciences Bremen, University of Bremen, 28359 Bremen, Germany; 5Unit of Health Promotion Research, University of Southern Denmark, 6705 Esbjerg, Denmark; 6Sociology of Health and Health Care Systems, University Siegen, 57076 Siegen, Germany; wendt@soziologie.uni-siegen.de

**Keywords:** COVID-19, student health, fear of infection, depressive symptoms

## Abstract

The COVID-19 pandemic has a significant psychological impact at the population level and fear of infection is one of the stressors involved. The study aimed to examine fear of infection and associations with university students’ depressive symptoms, substance use, and social contacts during the COVID-19 outbreak in Germany in May 2020. A cross-sectional online survey was conducted at four German universities (*n* = 5.021, 69% female, mean age: 24 years) as part of the COVID-19 International Student Well-being Study. Fear of infection was assessed using self-generated items, depressive symptoms were assessed using the Center of Epidemiologic Studies Scale (CES-D-8). Associations between fear of infection and depressive symptoms were analyzed with linear regressions, controlling for sociodemographic variables. A total of 34% of the participants reported feeling worried about getting infected themselves, 75% were worried about someone from their personal network getting infected, and 78% feared that individuals close to them would get severely ill after infection. Sixteen percent of the variance of depressive symptoms could be explained by fear of infection (*p* ≤ 0.001). Students’ fear of infection should be considered in student communication and counseling to prevent worsening of mental health in this population.

## 1. Introduction

Since the first case of the novel coronavirus disease (COVID-19) was reported in Wuhan, China, in December 2019, the SARS-CoV-2 virus has spread around the world and has led to a global public health crisis [1]. Researchers are warning about possible long-term psychological consequences of the pandemic, such as increasing suicide rates [2,3]. An increase in symptoms of anxiety, depression, post-traumatic stress disorder, psychological distress, and stress was reported in a recent publication [4]. Student status was identified as one of many risk factors for developing mental health problems during the pandemic, as well as female gender and/or younger age (<40 years) [4]. A literature review by Zürcher et al. (2020) [5] called attention to the increase in mental health problems during epidemics and stressed the need for the provision of mental health care to mitigate these problems [5].

The fear of infection with COVID-19 is a stressor that may occur as a direct consequence of the COVID-19 pandemic [6]. Previous studies investigated the fear of infection with COVID-19 and its impact on mental health and found associations with depression and anxiety among university students [7,8]. A newly developed “Fear of COVID-19 Scale” operationalizes these worries and fears about the coronavirus [7]. This scale was translated into different languages and adapted to the countries’ contexts [9]. The development of a scale that measures fear of COVID-19 highlights its relevance for studies that focus on the topic of mental well-being during the pandemic. 

SARS-CoV-2 is a virus with a high infection rate [10] and relatively high mortality rate, especially for older people and people with pre-existing conditions [11]. Fear of infection is naturally linked with infectious diseases. Particularly, the fear of getting in contact with a person infected with COVID-19 was reported in previous studies [12]. From prior epidemics, it is known that fear may lead to negative consequences affecting disease control [12]. For example, during the outbreak of Ebola in West Africa in 2014, the fear of infection with the virus led to rumors about the virus and, in some cases, denial of the virus [13].

Worries and concerns about getting infected can be observed in the current pandemic, even in populations with a low risk of infection because of a low prevalence and incidence in the geographical area that they are located in [14]. However, there is also evidence that the fear of infection with COVID-19 was higher in areas that reported many cases [15]. Further, it is known that fear of COVID-19 infection is prevalent among students in higher education [8,16]. While fear is a natural and protective reaction, it can have severe psychological consequences [13]. For example, prior research found an association between fear of COVID-19 infection (after exposure to an infected individual) and depressive symptoms among a sample of community-dwelling adults [17]. High levels of fear of COVID-19 infection were associated with higher levels of depression among university students [8]. Another study demonstrated that higher levels of fear of infection with COVID-19 were associated with higher levels of alcohol and tobacco use as well [18]. Levels of problematic alcohol use were particularly high in university students with high levels of stress [19] and anxiety and depression [20]. COVID-19-related worries and fears were found to encourage substance use, especially the initiation of substance use during the pandemic [21]. Thus, current studies suggest that students’ mental well-being is at risk during the pandemic [22,23] and research should continue to focus on the vulnerable population of university students. 

Further, physical distancing appeared to have a negative impact on mental well-being during the ongoing pandemic [24,25], as social isolation in times of COVID-19 was associated with loneliness [26]. The fact that universities were closed and students had to get used to online teaching may have reduced students’ social contacts and increased isolation. Particularly students with higher levels of fear of COVID-19 infection were found to reduce social contacts in previous studies [12,27]. Taking the prevalence of depressive disorders in student populations prior to the pandemic into consideration [28], it is conceivable that the measures, including social distancing and self-isolation, may have had a negative impact on students’ mental well-being, as previously suggested by Karasmanaki and Tsantopoulos [29].

While there is increasing evidence of international studies [6,16,17,23,30] regarding the impact of COVID-19 on depressive symptoms, substance use behavior, and social behavior in the general population and international student populations, including the frequency of social contacts and activities, little is known about the interplay of these factors in university students in Germany. Thus, this study investigated the fear of infection with COVID-19 among German university students and associations with depressive symptoms, substance use, and social contacts. As the construct “fear of COVID-19” is new, one aim of the study was to describe different aspects of fear in this population. 

The following hypotheses were formulated based on international research on the topic, to date, and tested based on the data of a cross-sectional online survey conducted with 5021 university students at four different universities during the first wave of the COVID-19 pandemic: H1: Higher levels of fear of infection with COVID-19 predict higher levels of depressive symptoms among both female and male university students.H2: Female university students report higher levels of fear of infection with COVID-19 when compared with male university students.H3: Higher levels of fear of infection with COVID-19 are associated with higher levels of tobacco, alcohol, and cannabis use.H4: Higher levels of fear of infection with COVID-19 are associated with reduced social contacts with family and friends.H5: Higher levels of fear of infection with COVID-19 are associated with a lower number of offline activities, but a higher number of online activities.

## 2. Materials and Methods

### 2.1. Survey and Recruitment 

This study is part of the COVID-19 International Student Well-being Study (C19 ISWS) examining the impact of the COVID-19 pandemic on student well-being at 110 higher education institutions in 27 European and North American countries, as well as in South Africa, in 134,000 college and university student participants. A research team of the University of Antwerp (UAntwerp) planned and coordinated the international study. An online survey was used to collect data on the student populations. Ethical approvals for the study were obtained through the participating institutions. The aims and the methods are described in further detail in the study protocol [31].

Briefly, the survey questions were originally developed by the consortium leaders, and questions were subsequently refined in the international consortium. Each country had the possibility to add questions to the survey. In line with the translation protocol, the survey was translated independently by two authors of the German team (Stefanie Helmer and Heide Busse). The translation was discussed, in detail, and any disagreements were resolved by reaching a consensus. The questionnaire was translated back into English for this article. The translated questions were inserted into the Qualtrics software (Qualtrics LLC, Provo, Utah, USA). Data collection, using the online Qualtrics survey, was carried out from 12 May to 29 May 2020. The survey was distributed at each German university by inviting students to participate in the study via university websites, mailing lists, and social media accounts of the universities. The student associations at the universities also helped promote participation in the survey among students. The survey ended after a period of two weeks and UAntwerp returned the fully anonymized dataset after data cleaning to the partner institutions within another two weeks. The core questionnaire is publicly available [31]. 

### 2.2. Study Sample and Context 

A total of 5021 students were recruited at the following German universities: Charité-Universitätsmedizin Berlin, University of Bremen, University of Siegen, and Heinrich Heine University Duesseldorf. The aim to sample at least 10% of students was achieved in three of the four participating German universities. The actual rates of return have been reported in a previous article [32]. The approximated response rates for three of the universities were 10–11%. In the university that solely recruited via email in one of five faculties (Medical Faculty) and which used social media for recruitment of the overall student population at the university, a response rate of approximately 2% was reached at the university level, and a response rate of approximately 17% at the faculty level. Approximately two thirds of the sample came from the University of Bremen and the University of Siegen and the other third consisted of students from the Charité-Universitätsmedizin Berlin and the Heinrich Heine University Duesseldorf. At the time of data collection, the first lockdown in Germany that lasted from 22 March to 6 May had just ended. During the first strict lockdown, universities and schools closed and people were only allowed to leave the house to go to work, if necessary, to go grocery shopping, to go see the doctor, or to go to another very important appointment and to do sports outside or to go for a walk [33]. The universities in Germany predominantly remained closed until today. The majority of lectures and classes continued to be held online.

The preliminary findings of this article were presented at the annual conference of the International Congress of Behavioral Medicine (ICBM) in June 2021. The abstract of this poster presentation is included in the abstract book, but does not include the detailed analysis presented in the current article [34].

### 2.3. Measures

The online survey includes various questionnaires in order to investigate students’ well-being during the lockdown. In addition to sociodemographic characteristics, such as age, gender, nationality, field of study, financial status, and living situation of students, the survey included variables to capture health behavior, such as substance use and physical activity. COVID-19-related information, e.g., whether students had been previously infected with COVID-19 and the type of symptoms, were also assessed. Additionally, perceived study conditions and social contacts were assessed in the survey. For this investigation, the variables “fear of COVID-19 infection”, “depressive symptoms”, “social activities”, and “substance use” are relevant and will therefore be described in further detail below. 

#### 2.3.1. Fear of COVID19 Infection

In order to measure the “fear of infection with COVID-19”, students had to answer the following questions concerning their worries about a possible infection for themselves or others: (1) In your opinion, how likely are you to get infected with COVID-19?; (2) How worried are you to get infected with COVID-19?; (3) How worried are you to get severely ill from a COVID-19 infection?; (4) How worried are you that anyone from your personal network will get infected with COVID-19?; (5) How worried are you that anyone from your personal network will get severely ill from infection with COVID-19?; and (6) How worried are you that doctors and hospitals will not have sufficient medical supplies to handle the COVID-19 outbreak? Response options were presented on a scale from 0–10, ranging from not worried to very worried, or from not likely to very likely for item number one. For an easier interpretation of the scores, we formed four categories, for each item: Scores 0–1 were categorized as “not worried”, 2–4 as “little worried”, 5–7 as “worried”, and scores above 7 formed the category “very worried”. The fear of COVID-19 scale consisted of 6 items (α = 0.79). 

The sum score was used to analyze the fear of infection among students following the approach of Tasso et al. [35]. The fear of infection scale sum score can range from 0 to 60 (M = 29; SD = 11.5). A higher sum score represents greater fear of infection for the students themselves and their loved ones. Sum scores were categorized based on the quartiles. A sum score from 0–20 represents category one: “not worried”, scores from 22–29 form category two: “little worried”, scores from 30–37 form category three: “worried”, and students with a score above 37 were categorized as being “very worried”.

#### 2.3.2. Subjective Depressive Symptoms

Subjective depressive symptoms were assessed using the short version of the Center for Epidemiological Studies Depression Scale (CES-D 8) [36]. This scale assesses how much of the time during the last week students (1) felt depressed, (2) felt that everything was an effort, (3) their sleep was restless, (4) felt happy, (5) felt lonely, (6) enjoyed life, (7) felt sad, and (8) felt like they could not get going. Items (4) and (6) were reversely coded. A detailed description of the use of the CES-D-8 scale in our student sample can be found in Matos Fialho et al. (2021) [37]. For this investigation, a score >9 on the CESD-8 scale was used to define depression, as previously proposed by Briggs et al. (2018) [38]. Cronbach’s alpha for the 8 items was α = 0.85. The long version of the Center of Epidemiological Studies Depression Scale was validated for the German population by Stein et al. (2014) [39]. The response options were presented on a four-point Likert scale ranging from (0) none or almost none of the time, (1) some of the time, (2) most of the time, to (3) all or almost all of the time. Responses to the individual items were summed up to create the overall CESD-8 score. The sum score can range from 0 to 24, a higher score suggesting higher levels of depressive symptoms.

#### 2.3.3. Substance Use

Substance use items cover students’ drinking behavior, cigarette smoking, and cannabis use. The items of interest were: (1) number of cigarettes smoked, (2) number of drinks consumed, (3) binge drinking, (4) smoking behavior, and (5) cannabis use. Substance use among students was assessed for two time points: retrospectively, “before the pandemic”, and at the time of data collection “during the last week”. 

(1)Students were requested to indicate the number of cigarettes they smoked per day prior to and during the outbreak of COVID-19. Smoking was categorized as “none”, “1–9 cigarettes per day”, and “10 or more cigarettes per day”.(2)Students had to indicate the number of drinks they had had per week prior and during the outbreak. One drink was defined as one glass of an alcoholic beverage, such as beer or wine of 25 to 33 centiliters. The number of drinks was categorized into “none”, “1–2 drinks per week”, “3–5 drinks per week”, and “5 or more drinks per week”.(3)Binge drinking defines the consumption of six glasses of alcohol or more on one occasion [40]. Students were asked to report how often during the last week and before the emergence of the pandemic they had had six or more drinks on one occasion. The response options were coded on a five-point Likert scale, ranging from (1) “(almost) never” to (5) “(almost) daily”. The response options for the frequency of binge drinking were grouped into four categories: “never” and “less than once per week” were grouped as one category, while the other categories remained the same, i.e., “once every week”, “more than once per week” and “(almost) daily”.(4)In addition to the number of cigarettes consumed, the frequency of cigarette use was assessed with the following item: “On average how often did you smoke tobacco (cigarettes, e-cigarettes, or cigars) before the pandemic/during the last week”?(5)The third behavior that was assessed was cannabis use. Students were asked: “On average, how often did you use cannabis (marijuana, weed, or hash) before the pandemic/during the last week?” The response options were categorized the same way as for binge drinking and cigarette use.

#### 2.3.4. Social Activities

Students were asked the following question: “During the last week did you engage in one of the following activities?” There were ten possible activities listed and the response options were binary (i.e., yes/no). The following activities were listed: (1) walk, (2) bike ride, (3) drink or picnic, (4) talked to friends or family on the street, (5) recreational class online, (6) game or quiz online with friends or family, (7) talked to friends or family through a video call, (8) talked to friends or family over the phone, and (9) chatted with friends and family online. We added the response option “None of the above” as well. The activities were summed up with a high sum score indicating that the student engaged in many activities. The scale to measure social activities during the first wave of COVID-19 in Germany consisted of 9 items (α = 0.54). A sum score of offline activities was created to identify possible differences in the kind of activities students engaged in, depending on the fear of infection. This sum score included the activities: “walk, bike ride, drink, or picnic and talked to family and friends on the street”. This score ranged from (0) none of these activities to (4) all of these activities. The score of online activities included the activities: “recreational class online”, “game or quiz online with friends or family”, “talk to friends or family through a video call”, “talk to friends or family over the phone”, “chatted with friends or family online”. In order to assess increases or decreases in social contacts, students were asked the following questions: “Did you have more or less contact (online and offline combined) with family since the implementation of the first COVID-19 measures?” and “Did you have more or less contact (online and offline combined) with your friends since the implementation of the first COVID-19 measures?” The response options were presented on a 3-point scale ranging from (1) “More”, (2) “About the same”, to (3) “Less”.

### 2.4. Covariates

The following covariates were included in the analyses: Gender (female/male/diverse), age (17–19, 20–23, 24–27, 27–30, and 31 years and older), relationship status (single/in a relationship/it is complicated), resident status in Germany (permanent residency/temporary residency), availability of a person to discuss intimate matters with (yes/no), level of education of both parents (less than secondary education, secondary education, higher education, or I do not know).

Students were asked which study program they were currently enrolled in (Bachelor’s program/Master’s program/doctoral program/state examination (medicine, law/other). They were also asked, if it was their first year in higher education (yes/no) and what best described their field of study (education, arts, humanities, languages, social and behavioral sciences, journalism, media and communication, business and administration, law, natural sciences and mathematics and statistics, etc.). Regarding their current living situation, students were asked where they mainly lived during the initial COVID-19 outbreak (excluding weekends and holidays). The response options were the following: with parents, student hall, accommodation with others, accommodation alone and other. Concerning their financial situation, students were asked to indicate whether they had sufficient financial resources to cover their monthly expenses or not during the COVID-19 outbreak. Regarding COVID-19, the two covariates were whether students had already been infected with COVID-19 or whether they knew someone in their personal network who had already been infected and how severe the disease was among the people they knew who had been infected. 

### 2.5. Data Analysis

To evaluate the sociodemographic characteristics of our study sample, we performed descriptive analyses. For the fear of infection, substance use, social activities, and depressive symptoms during the COVID-19 outbreak, absolute (*n*) and relative (%) frequencies were calculated. 

Associations between fear of infection (predictor variable) and depressive symptoms (dependent variable) were analyzed with linear regressions, controlling for sociodemographic variables. The variables: fear of infection, age, gender, relationship status, parental education, field of study, study program, first year of education, living situation, financial resources, resident status, and availability of a person to discuss intimate matters with were included as independent variables in the model. 

Univariate variance analysis was used to identify mean differences in the fear of infection (dependent variable) among the students in association with their different levels of substance use (factor). We ran this analysis for all five aspects of substance use: (1) number of cigarettes smoked, (2) number of drinks, (3) binge drinking, (4) smoking frequency, and (5) cannabis use. When the ANOVA yielded statistically significant results, post hoc t-tests were undertaken to identify mean differences between substance use categories concerning the levels of fear of infection. To investigate possible associations between the number of social activities students engaged in and their fear of infection, we calculated bivariate correlations. We ran correlations for fear of infection and offline activities and online activities separately. In order to assess perceived frequency of contact with family and friends since the outbreak and the association with fear of infection, two bivariate correlation analyses were performed, one for the frequency of contact with family and one for contact with friends. Data analysis was performed using IBM^®^ SPSS^®^ version 26 (IBM, Armonk, NY, USA).

## 3. Results

### 3.1. Characteristics of the Participants 

The baseline characteristics of the sample have been previously reported in detail [37]. Briefly, over 70% of the students were female and the median age was 23 years (mean age: 24.4 years (SD 4.5)). In terms of relationship status, 53% reported being in a steady relationship. About 53% of the students were enrolled in a Bachelor’s program and over 20% in a Master’s program. Twenty-five percent of the students were medicine or health sciences students. The majority of the participating students were enrolled in a program at Bremen University (37.6%) and at Siegen University (34.4%). Twenty-three percent of the students were first year students at the time of data collection. 

Regarding COVID-19 infection, approximately 91% of the participants reported that they had not had a positive diagnosis for SARS-CoV-2, while 8% of the students stated that they thought they had already been infected, despite a lack of medical evidence based on a lab test. Thirty-two percent of participants reported knowing someone in their personal network who was infected with COVID-19 at the time of data collection or prior to that. In most reported cases, the symptoms of the infected person were mild or severe and not requiring intensive care. Nevertheless, about 3% reported that the infected person was deceased.

Half of the students in our sample reported that they did not have sufficient financial resources to cover their monthly costs. Around 10% of the participants reported that they did not have anyone to discuss personal and intimate matters with. Six percent of the sample did not have a permanent resident status in Germany. Regarding the education level of the students’ parents, 42% of the students’ fathers obtained higher education and 33.5% percent of the students’ mothers. Forty percent of the students reported that their mother obtained less than secondary education, as well as 37% of the participants’ fathers. At the time of the data collection, the majority of the students (33.3%) lived with their parents or in an accommodation with others (25%).

### 3.2. Fear of Infection with COVID-19

Table 1 provides a detailed overview of the descriptive results regarding fear of COVID-19 infection among study participants. The fear of getting infected with COVID-19 seemed to be present in the entire student population. Approximately half of the students reported infection to be likely or even very likely, the other half thought that it was very unlikely or unlikely. Around 24% of the students worried about possible infection with the new virus and 10% of the students reported feeling very worried about infection. While only 35% of participants reported being worried or very worried to personally get infected, 75% felt worried concerning infection of someone in their personal network. Moreover, students were more worried about a severe progression of the disease for others than for themselves. Regarding the trust in the health care system, 47.4% of the students felt worried regarding a possible overload of the healthcare system, while the other 52.6% were not worried that doctors or hospitals could lack medical supplies to handle an outbreak. As predicted, there was a statistically significant difference between levels of fear of COVID-19 infection for female and male students, with higher mean levels of fear among female students (t(4917) = −10.67, *p* <0.001). 

### 3.3. Depressive Symptoms

The results regarding subjective depressive symptoms of students during the outbreak of COVID-19 can be found in Table 2. One in four students felt depressed almost all of the time or most of the time at the time of data collection. Thirty percent of the participants felt that they could not get going. The mean sum score of the CES-D 8 scale was M = 9.25 (and hence above the cutoff of 9) indicating that the students were slightly depressed [38].

### 3.4. Social Activities

Table 3 shows the descriptive results regarding social activities and social contacts. Fifty-seven percent of students reported having had less contact (offline and online combined) with friends since the implementation of the first COVID-19 measures. However, the frequency of contact with their families did not change since the COVID-19 outbreak for the majority of students (49%). 

### 3.5. Regression Model for Fear of Infection and Depressive Symptoms

In the linear regression, adjusted for the selected covariates (see Table 4), 16% of the variance of depressive symptoms could be predicted by fear of infection and the other covariates (R^2^ = 0.16, F(13, 4916) = 73.60, *p* <0.001). Female gender was associated with higher levels of depressive symptoms. Those students that did not have sufficient financial resources to cover their monthly costs were more likely to report higher scores of depressive symptoms. Moreover, students in a steady relationship (compared with being single) and those indicating the presence of a person to discuss personal matters with (compared with not having such a person in their lives) were less likely to report depressive symptoms. Students without a permanent residence status were more likely to report depressive symptoms than those with a permanent residence status. 

### 3.6. Associations between Substance Use and Fear of Infection

Table 5 shows the results of the ANOVAs analyzing associations between different levels of substance use and fear of infection. There was an association between the frequency of tobacco use and the fear of infection (higher levels of fear were associated with higher levels of tobacco use; F(3, 4.915) = 3.39, *p* < 0.05). There were no significant differences in the levels of fear of COVID-19 infection for the different levels of binge drinking, number of drinks consumed, number of cigarettes smoked, and cannabis use. 

Post-hoc analyses demonstrated that the differences in the mean levels of fear of infection were statistically significantly elevated for those who never smoked (or less than once a week) as compared with those who smoked (almost) daily (t (4699) = −3.02, *p* < 0.05). 

### 3.7. Associations between Social Activities and Fear of COVID-19 Infection

Contrary to the hypothesis that students with higher levels of fear of infection with COVID-19 would report to have less contact with their families, the perceived frequency of social contact with the family was not associated with students’ fear of infection (*p* = 0.783). It was also hypothesized that those with higher levels of fear of COVID-19 would report less contact with their friends. Even though the contact with friends decreased for the majority of students, no significant correlation with their level of fear of infection was found (*p* = 0.135). Regarding the number of social offline activities, an association was found. Students with higher levels of fear reported fewer offline-social activities during the last week (r = −0.077, *p* < 0.001). A low but significant positive correlation between the number of online activities and the fear of COVID-19 among the students was found as well. Students with higher levels of fear reported engaging in many online activities (r = 0.081, *p* < 0.001). 

## 4. Discussion

This article investigated the fear of infection with COVID-19 and its associations with depressive symptoms of students during the ongoing COVID-19 pandemic. We found that a small group of students was worried to get infected with COVID-19, but it seems that the fear for someone in their personal network to get infected and to become severely ill was more pronounced among students. These results confirm the findings of a previous study on student mental well-being during the COVID-19 pandemic in the United States of America [23]. This study assessed COVID-19-specific worries in 18,764 students during the pandemic. In this sample, 64.4% of the students reported being extremely worried about people they cared about contracting COVID-19 and 40.9% also met the criteria for clinical depression [23]. Thus, students appeared to be more worried about their loved ones than about themselves. COVID-19 is a virus especially threatening to a group of people at an advanced age (>65 years) and people with pre-existing conditions, such as obesity and diabetes [41]. Therefore, students in our study might not have considered themselves as part of a high-risk group.

Further, higher levels of fear of COVID-19 infection were found to be associated with higher levels of depressive symptoms among students in our study. This finding is in accordance with the results of a previous study that suggested an association between the fear of infection with COVID-19 and depression and anxiety among university students [7,8,16]. One dimension of the fear of COVID-19 scale in this study is the trust in the healthcare system. This trust could be a protective factor concerning mental well-being in the pandemic. Students who felt confident that physicians and hospitals would have sufficient supplies to handle a COVID-19 outbreak were less afraid of infection and, therefore, had lower levels of depressive symptoms. Similarly, previous studies found that trust in the healthcare system alleviated the fear of the consequences of a hypothetical infection [14].

Female students reported higher levels of fear of the virus when compared with male students. This was already reported in a prior study among university students from Russia and Belarus [18]. Female students might be more aware of potential health threats and the fear of infection with COVID-19 may be more present among female students or they may be more likely to report it compared with male students.

We hypothesized that higher levels of fear of infection with COVID-19 would predict higher levels of depressive symptoms (H1). Our results could confirm this hypothesis. The fear of infection with COVID-19 was found to be a stressor for students’ mental health. Moreover, similar to other literature [18], we assumed that female students would report higher levels of fear of infection with COVID-19 when compared with male students (H2). Again, our findings confirmed the findings of previous studies [18,42].

In addition, this study investigated the associations between fear of COVID-19 infection and alcohol, tobacco, and cannabis use among students during the pandemic. We hypothesized that higher levels of fear of COVID-19 infection would be associated with higher levels of substance use among the students (H3). An association between the frequency of tobacco use and fear of COVID-19 infection was found (H3). However, the hypothesis could not be confirmed for alcohol and cannabis use. 

Further, this study examined the association between the fear of infection with COVID-19 and the frequency and type of social contacts of students during the lockdown. Our hypothesis was that students with higher levels of fear of infection with COVID-19 would report a reduction in contact with family and friends and a lower number of social activities (H4). Based on our findings, this hypothesis must be rejected. However, we also hypothesized that students’ levels of fear of COVID-19 would be associated with the number of social activities they engage in online and offline (H5). This assumption was supported by our findings. Higher levels of fear of COVID-19 were associated with fewer social offline activities and more social online activities. 

Furthermore, we found that students who smoked (almost) daily during the week preceding data collection displayed a higher mean score for fear of COVID-19 infection when compared with non-smokers. One possible interpretation is that students who were more afraid were smoking cigarettes more often to cope with their anxiety, as smoking cigarettes is often used as a coping strategy [43]. On the other hand, a previous study found that cigarette and alcohol use had a negative impact on the fear of COVID-19 infection in a comparable population of college students [42]. Smokers may have also felt a greater fear of infection because of a perceived tobacco-induced respiratory vulnerability to COVID-19. It seems that use and fear of infection are associated, but based on our results, no inferences regarding causality can be made.

Another hypothesis in this study was that students with higher levels of fear of infection would report a decrease in social contacts with family and friends since the implementation of the COVID-19 measures. Although the majority of the students (58%) reported having had less contact with their friends since the outbreak, decreased contact was not associated with their COVID-19-related fear. COVID-19 restrictions may have mainly affected social “offline” contacts. Students may have reduced their “in-person” meetings with friends and families, but the contact might have predominantly taken place online. It should be noted, however, that it is a limitation of this study that the question used in the survey to assess social contacts did not differentiate between online and offline contact with family and friends.

Students who felt more afraid to get infected with the virus and/or to infect others, may have shifted their contact to online meetings. The leisure-time activities that students engaged in during the last week preceding data collection were also investigated. Regarding the offline activities, the expected correlation with the fear of COVID-19 infection was found: higher levels of fear of COVID-19 were associated with a lower number of offline activities. Students who were more afraid of infection with the virus or more afraid to infect others might have engaged in fewer social activities that were not held online. It might be reasonable to assume that the decrease in outside activities and meetings with family and friends offline promoted a sense of loneliness. During this pandemic, many activities (leisure and work-related) took place online in order to prevent the spreading of the virus and students were particularly affected by (the still ongoing) online teaching [44].

Worries and fears about possible infection with COVID-19 are significant stressors affecting students’ mental well-being. Universities should provide student counseling and mental health care available online and offline and easily accessible. Many researchers warn about a second pandemic of mental health problems [45] as a possible consequence of the COVID-19 pandemic and call for measures in order to minimize this effect [46].

### Limitations

Several limitations of this study should be noted. First, the data collection for this study took place in May 2020 in Germany. At the time the survey was conducted, the first lockdown in Germany had just ended, but many restrictions still remained. Nevertheless, there was a noticeable relief regarding the pandemic in the German population. The findings of this study are only a snapshot of the time of data collection and the results are extremely dependent on the current level of infections. Second, the construct “fear of infection with COVID-19” is still fairly new. Since the implementation of our survey in May 2020, several scales for assessing this fear were designed and validated, including the “Fear of COVID-19 Scale” [7], the “Coronavirus Anxiety Scale” [47], or the “Coronavirus Stress Scales” [48]. At the time our survey was developed, these scales were not available which is why the six items that capture the fear of infection with COVID-19 in this study were self-generated. For improved comparability of results and higher statistical power, upcoming studies should rather use one of the now available validated scales on fear of infection. Third, the students in the study sample reported being worried about someone in their personal network getting infected with COVID-19. However, the survey did not capture who the people in their personal network exactly were. As mentioned before, it can be assumed that students worried about those people in their personal network more threatened by the virus (e.g., with a pre-existing condition). About a quarter of the students in our sample were students enrolled in medicine or health sciences. It is known from prior research that health literacy is a protective factor for fear of COVID-19 among medical students [42]. Therefore, the low number of students who reported feeling worried about infection could be explained by higher levels of health literacy in this population. Further, possible selection bias needs to be considered in the interpretation of results, as students feeling affected by the pandemic might have preferably taken part in the survey. Finally, ad hoc retrospective and current substance use items could have been influenced by underreporting bias. The lack of association between substance use and fear of infection, except for tobacco use, might reflect this methodological issue.

## 5. Conclusions

To conclude, this study provided first insights into associations between fear of infection with COVID-19 and depressive symptoms among university students in Germany. The results of this study call for attention to students’ mental well-being during the COVID-19 pandemic. Universities, as well as governments, should consider in their decision making that students are psychologically burdened by the ongoing pandemic. The findings of this study and several recent other studies show the urgent need to develop preventive strategies addressing the mental health of university students in the current situation, as well as in future pandemics [22]. Future research should focus on potential long-term mental health consequences of the pandemic in student populations. 

## Figures and Tables

**Table 1 ijerph-19-01659-t001:** Descriptive baseline characteristics of fear of infection.

Item	*n*	%
In your opinion, how likely are you to get infected?		
Not likely	587	11.9
A little likely	1910	38.7
Likely	1849	37.4
Very likely	594	12.0
How worried are you to get infected with COVID-19?		
Not worried	1235	25.0
A little worried	2020	40.9
Worried	1213	24.6
Very worried	472	9.6
How worried are you to get severely ill from a COVID-19 infection?		
Not worried	2058	41.7
A little worried	1702	34.5
Worried	740	15.0
Very worried	440	8.9
How worried are you that anyone from your personal network will get infected with COVID-19?		
Not worried	270	5.4
A little worried	905	18.2
Worried	1710	34.4
Very worried	2083	41.9
How worried are you that anyone from your personal network will get severely ill from infection?		
Not worried	300	6.0
A little worried	734	14.8
Worried	1403	28.2
Very worried	2531	50.9
How worried are you that doctors and hospitals will not have sufficient medical supplies to handle the COVID-19 outbreak?		
Not worried	895	18.0
A little worried	1460	29.4
Worried	1378	27.7
Very worried	1235	24.9
Fear of infection with COVID-19 Mean (SD)	29 (11.5)	

**Table 2 ijerph-19-01659-t002:** Description of subjective depressive symptoms.

Items of CES-D 8 Scale	All or Almost All/Most of the Time		Some of the Time		Almost None or None of the Time	
	*n*	%	*n*	%	*n*	%
Felt depressed	1240	25.2	2541	51.5	1152	23.4
Felt that everything they did was an effort	1489	30.2	2046	41.5	1398	28.3
Sleep was restless	1447	29.3	2064	41.8	1422	28.8
Happy	2540	51.4	2060	41.8	333	6.8
Felt lonely	994	20.1	1916	38.8	2023	41
Enjoyed life	2105	42.7	2177	44.1	651	13.2
Felt sad	920	18.7	2683	54.4	1330	27
Could not get going	1440	29.1	2033	41.2	1460	29.6
Mean sum score (SD) 95% CI	9.25 (4.7) 9.11–9.38					

**Table 3 ijerph-19-01659-t003:** Descriptive results regarding social activities and social contacts.

Item	*n*	%
Engagement in sports or social activities during the last week		
Walk	3332	66.4
Bike ride	1183	23.6
Drink or picnic	2163	43.1
Talked to friends or family on the street	2322	46.2
Recreational class online	1551	30.9
Game or quiz online with friends or family	1833	36.5
Talked to friends or family through a video call	3614	72.0
Talked to friends or family over the phone	3937	78.4
Chatted with friends or family online	3314	66.0
None of the above	66	1.3
Contact offline and online combined with family since the implementation of the first COVID-19 measures		
More	1762	35.1
About the same	2440	48.6
Less	750	14.9
Contact offline and online combined with friends since the implementation of the first COVID-19 measures		
More	735	14.6
About the same	1367	27.2
Less	2850	56.8

**Table 4 ijerph-19-01659-t004:** Regression on depressive symptoms with the predictor fear of infection with COVID-19.

	Regression on Depressive Symptoms	
Variable	Regression Coefficient (SD)	*p*
Intercept	0.25(0.66)	0.703
Fear of infection	0.071 (.005)	0.000 **
Age	−0.24 (0.14)	0.082
Gender	0.973 (0.13)	0.000 **
Field of study	−0.029 (0.009)	0.002 *
Study program	−0.446 (0.05)	0.000 **
First year of education	−0.077 (0.15)	0.609
Relationship status	−0.394 (0.11)	0.001 *
Residence status	0.89. (0.27)	0.001 *
Highest level of education mother	−0.04 (0.08)	0.643
Highest level of education father	0.008 (0.07)	0.913
Living situation	0.158 (0.06)	0.007 *
Financial resources	2.33 (0.18)	0.000 **
Person to discuss intimate matters with	3.53 (0.22)	0.000 **

Note: *n* = 4.930, significant associations: * *p* < 0.05. ** *p* < 0.001.

**Table 5 ijerph-19-01659-t005:** Between-groups mean comparisons of fear of COVID-19 infection by substance use.

Substances	Sum of Squares	Mean Square	F	*p*
Frequency of cigarette use	1338.1	446.0	3.39	0.017 *
Binge drinking	334.5	111.5	0.84	0.469
Cannabis use	426.7	142.2	1.11	0.356
Number of cigarettes	204.7	204.7	1.49	0.223
Number of drinks	293.5	146.7	1.11	0.329

Note: *n* = 5.021, significant associations: * *p* < 0.05.

## Data Availability

Due to the nature of this research, participants of this study did not agree for their data to be shared publicly, so supporting data are not publicly available. Data are available on request from the corresponding author for collaborating researchers within the C19 ISWS consortium, as consent for this was provided from all participants.

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
