# Peer review of "Fear of Infection and Depressive Symptoms among German University Students during the COVID-19 Pandemic: Results of COVID-19 International Student Well-Being Study"

_ijerph, 2022, doi:10.3390/ijerph19031659_

Round 1
Reviewer 1 Report
This paper is based on data collected in May 2020. While specific to German university students, findings should be linked to prevailing conditions that are much worse than when the data were collected.
Furthermore, the results, tables, and more evidence a paper not acceptable for publication . The authors should consider a brief reserach report or letter to the editor to present key findings that may be relevant to prevailing COVID-19 conditions in Germany.
Suggested edits:
- 2 - first line and/or young age
- 2 - SARS-CoV-2 ..should be a new paragraph
- 2 - delete "in Ecuador"
- 2 - 2nd paragraph - ...especially its initiation
- 2 - current outbreak ? this study was done in May, 2020 (see p. 3)
- 3 - 2.1 - was the questionnaire back translated?
- 3 - 2.2 5,021 students were recruited - what was the actual rate of return?
- 4 - 2.3.1 title - check punctuation; Fear of COVID-19 scale, there should be a Cronbach alpha score reported. If the scale is to measure fear, why where the questions worded with "worry"? Values for the scale should be reported (mean; SD, medican, range, etc.; and, a distribution of results on the "fear/worry scale).
- 4 - 2.3.2 need to report Cronbach alpha
- 5 - 2.3.3 - this part needs to be rewritten with details about smoking behavior. A description of variables should not be left to Busse et al., 2021
- 5 = 2.3.4 - Cronbach alpha needs to be reported
Results, p. 6 - 50% were aged 20-23 years is inadequate; report exact median age.
- 6 & 7 - 3,2 - it would be helpful to add a sentence or two on the meaning of "worry "and why the word was used instead of "fear" in most, "Fear and worry" are used interchangeably. The authors mention there are statistically significant differences between levels of fear ...among female and male students; however, there are no fear values reported. Also, the purpose of Table 1 is questionable - it lacks information about students based on different faculties, age, gender, work, marital status, etc.. same for table 2, 3 and 4 (see 3.3) ...there is no reporting of statistical analyses based on these factors.
3.5 - Table 5 (Regression) .It is not clear why regression analysis is needed to confirm H1 . The results are not used in the paper and are not explained. ...there are too many variables used for the analysis and the authors should include the R square value.
3.6 Table 6 - there is too little data provided about substance use- see comment above for Tables 2, 3, ....
- Discussion. edit/ revise. Overall, the paper should be rejected.
The data are dated and provide little insight of prevailing conditions since the study was conducted in May 2020.
Reviewer 2 Report
The study aim is to examine fear of COVID-19 infection and its relations with mental well-being, substance use, and social activities in a large sample of students in Germany. This cross-sectional survey is part of a larger study and addresses important research questions. The manuscript is well-written.
However, I have some comments.
A description of restrictions, lockdowns or stay-at-home orders in Germany during tha administration of the survey should be useful to understand the context.
“scores above 8 formed the category “very worried”.” Should be above 7.
As the Authors do acknowledge, the item “Engagement in sports or social activities during the last week” should have been presented in the Methods dividing talking on the phone or through a video call or chatting online from offline activities. In fact, in the Results section, they are analyzed separately and it is correct. Moreover, online sociality can be considered a reaction to physical distancing.
In the regression model on poor mental well-being with the predictor fear of infection, depressive symptoms and loneliness are considered together. Usually, distress and mental health well-being are measured with depressive and anxiety symptom scores, while loneliness is potential predictor and an associated variable, but it is a different construct. The Authors should clarify why subjective mental well-being measure included loneliness.
Ad hoc retrospective and current substance use items could have been influenced by underreporting bias. The lack of association between substance use and fear of infection, except for tobacco, might reflect this methodological issue. This point should be addressed in the Limitation section.
Smokers reported higher fear of COVID-19 infection as compared to non-smokers. Among possible explanations, a perceived tobacco-induced respiratory vulnerability to COVID-19 might be considered.
Reviewer 3 Report
The article brings some news regarding the relationship between the fear of COVID-19 and mental health, especially since the research is conducted on a large sample of students. Consider that an important point is that a scale built by the authors was used to measure the fears towards COVID. The studies that have used the well-known FCV-19 scale are in large numbers.
I have two recommendations:
If CES-D8 is validated on the German population, it is essential to reproduce this aspect (authors, year).
Secondly, other data extracted from the research on the reliability of the measures should be presented, especially for the two items in the ULS-8 scale (consistency coefficients or composite reliability, for example).
I found that the study still has a relatively similar title and the same authors in International Journal of Behavioral Medicine. In this case, the authors should indicate whether the present study is an extension of the IJBM article or is completely new.
Fear of infection and mental well-being among German university students during the COVID-19 pandemic Spatafora, F.; Matos-Fialho, P.; Busse, H.; Zeeb, H.; Helmer, S. M.; Stock, C.; Wendt, C.; Pischke, C. R.. International Journal of Behavioral Medicine ; 28(SUPPL 1):S15-S16, 2021. https://pesquisa.bvsalud.org/global-literature-on-novel-coronavirus-2019-ncov/resource/en/covidwho-1283135
